# CBOL: Cross-Bank Over-Loan Prevention, Revisited

**DOI:** 10.3390/e22060619

**Published:** 2020-06-03

**Authors:** Xiaoya Hu, Hong Zhao, Shihui Zheng, Licheng Wang

**Affiliations:** 1State Key Laboratory of Networking and Switching Technology, Beijing University of Posts and Telecommunications, Beijing 100876, China; huxiaoya@bupt.edu.cn (X.H.); marcus.zhao@tron.network (H.Z.); shihuizh@bupt.edu.cn (S.Z.); 2Technical Department, Golden Siv Technology Limited, Level 7, K11 ATELIER Victoria Dockside, 18 Salisbury Rd, Tsim Sha Tsui, Hong Kong 999077, China

**Keywords:** blockchian technology, privacy protection, over-loan prevention, amount hiding

## Abstract

With the development of credit businesses, privacy data leakage and data accuracy in loan transactions among different banks tend to be worrisome issues hindering the prosperity of the industry. To address the problem, we propose a blockchain-based cross-bank over-loan prevention (CBOL-ring) mechanism, which ensures that, on the one hand, the plaintext of loan transactions cannot be access to neither participants on the nodes except the bank that handles loan/repayment requests, so as to prevent the borrower from loaning without revealing their privacy data; on the other hand, the other participants are able to prove the effectiveness of the plaintexts through checking the ciphertexts on the blockchain. In addition, we propose a blockchain-based cross-bank over-loan prevention mechanism with low communication volume (CBOL-bullet), which reduces the size of the range proof generated by the BBCBOLP mechanism, thereby reducing the size of the communication volume and saving resources during the data transmission process. Finally, we analyze the security and performance of the two mechanisms, and compare the communication volume of the two mechanisms.

## 1. Introduction

In the credit business, the key issue is to propose a reasonable amount of loan that would not be beyond a borrower’s repayment capability. During the loan approval process, the bank would conduct a credit investigation on the borrower, and then generate a credit report, which includes the borrower’s loan records in other banks, financial status etc., to calculate the amount of the loan the borrower can apply for and make sure it would not exceed the repayment ability of the borrower.

Before the appearance of the credit investigation industry, banks mainly conducted credit investigation on borrowers in the following three ways: visiting the borrower, consulting other banks, reviewing previous loan records in local database, etc. [1]. Yet, this has some assicated problems.
Data privacy protection issues. The loan/repayment records of borrowers at a bank are not only part of the privacy data of the borrowers, but also the business assets of the bank. Protecting the borrower’s privacy data are a concern of the banking industry, but if the bank directly shares the loan/repayment data with other banks, the privacy of the bank and the borrower will be disclosed.Data security issues. In traditional ways, the borrower’s data are stored in a central server, which might expose all the data to hackers. Once the hacker successfully invades the database, the borrower’s private data are leaked, and on the other hand, the hacker can maliciously modify or delete the user information, so as to interfere with the bank to make the right decision on the borrower’s repayment ability.Data accuracy issues. The borrower’s data are invisible assets to a bank. The monopolization of a borrower’s information would improve the competitiveness of a bank, but the inter-bank data sharing would undermine the monopolization. Therefore, to improve the competitiveness, the bank might share the wrong information with peers [2].

With the emergence of the credit industry, the credit data of the borrower are collected by the credit reporting center, so are their data kept in banks. When a borrower applies for a loan in a bank, the credit reporting center would generate a credit report, on which the loan information of the borrower in other banks appears in plaintext, which definitely leaks the borrower’s privacy to the other banks. On the other hand, it confronts data with the risk of being modified or deleted in the process of transmission and storage.

In 2008, Nakamoto [3] first proposed the concept of blockchain, which has attracted worldwide attention. In addition to the research on blockchain technology itself, researchers are actively exploring the application of blockchain in a wide range of industries. Blockchain technology is an unforgeable, non-tamperable and traceable data structure built in a peer-to-peer (P2P) network. It is a distributed data storage system that uses P2P networks to propagate transactions and uses cryptography to connect blocks in the network; some other techniques such as merkle tree, time stamp, and smart contract are also applied on blockchain. Blockchain, as a distributed ledger, prevents data in the ledger from being tampered and does not require a trusted third party to prevent double-spending. It is maintained together by a lot of nodes to avoid single points of failure, and the data kept on the blockchain can be accessed by all nodes to reduce additional communication.

Cryptographic technology plays an important role in data protection. For example, encryption algorithms guarantee the confidentiality of data and prevent data from being modified during transmission; one-way hash function (hash algorithm) can guarantee the integrity of data; message authentication code can guarantee the integrity and authenticity of the data; data signature can guarantee the integrity, authenticity and the non-repudiation of the data. With the continuous improvement and development of cryptographic algorithms, cryptographic technology has been applied in various fields.

Based on blockchain and cryptographic technology, this paper constructs a mechanism to avoid the borrower applying the amount of loan in excess of his/her repayment ability from different banks, on which all the verified data are uploaded and encrypted, so that the loan information can be shared with all the participants on a chain and cannot be maliciously modified. This paper uses the Bulletproofs algorithm to optimize the communication volume of the proposed mechanism.

### 1.1. Contribution

Proposing a blockchain-based cross-bank over-loan prevention (CBOL-ring) mechanism. First of all, the mechanism uses commitment, public key encryption, ring signature, etc., which can not only solve the problem of privacy leakage in the process of data transmission and sharing, but also realize the co-verification of data without leaking sensitive information. Secondly, the mechanism uses blockchain technology to effectively prevent data from being modified or deleted. The involved banks can directly access to relevant data based on the openness of blockchain, thereby reducing the communications between banks and credit reporting center.Proposing a blockchain-based cross-bank over-loan prevention mechanism with low communication volume (CBOL-bullet). In this paper, the Bulletproofs algorithm is used to further improve the CBOL-ring mechanism proposed above to reduces the size of the range proof generated by the CBOL-ring mechanism, thereby reducing the size of the communication volume and saving transmission resources during the data transmission process.Evaluating the feasibility and security of the two mechanisms mentioned above through experiments.

### 1.2. Organization

The remaining part of this paper is organized as follows. In Section 2, we analyze some related works. In Section 3, we review a few terminologies and algorithms used in the work. In Section 4, we propose a system structure and adversarial model. In Section 5, we propose the CBOL-ring mechanism. In Section 6, we propose the CBOL-bullet. Section 7 is security and performance analysis. A conclusion is drawn in Section 8.

## 2. Related Work

Since blockchain has been proposed, the characteristics of decentralization, openness, transparency, and non-tampering have become hot research topics. With the development of blockchain technology, the application of blockchain tends to be extensive. As a result, more attention has been given to the issue of privacy protection. Although Bitcoin proposed by Nakamoto uses pseudonymous addresses to protect privacy, through analysis of transactions, people can link multiple addresses together, and the transaction amount and address of Bitcoin are publicly visible, consequently, the Bitcoin system does not achieve confidentiality. In order to protect the privacy of information on the blockchain, researchers have made more efforts in the recent years.

Some cryptographic techniques (such as Pederson Commitment [4], ring signature [5], zero-knowledge proof [6], etc.) are used in a blockchain. In 2013, Miers et al. [7] proposed Zerocoin, a distributed e-cash system. The system uses non-interactive zero-knowledge (NIZK) to hide the sender’s address, but the recipient’s address and amount transferred are not hidden. Through this scheme, individual Bitcoin transactions are not linkable, and it does not introduce a trusted third party, compared to other e-cash schemes. In 2014, Sasson et al. [8] proposed Zerocash, a decentralized anonymous payment currency scheme based on Zerocoin. In Zerocash, a zero-knowledge Succinct Non-interactive Arguments of Knowledge (zk-SNARK) is used to hide the sender’s address, transaction amount, and recipient’s address for privacy protection. In order to keep the amount confidential, in 2015, Maxwell [9] proposed the concept of confidential transactions, which refers to a cryptographic tools used to strengthen privacy in Bitcoin. Through confidential transactions, it can achieve the verification of the amount without revealing the amount (i.e., the encrypted state) and cannot create or destroy any Bitcoin. In 2015, the RingCT protocol was proposed by Noether [10] and used for confidential transactions in Monero. The ring signature protocol that forms the RingCT protocol was used to hide traders and transactions, but the protocol comes at the expense of the size of the transaction. In 2016, Jedusor [11] proposed MimbleWimble protocol. The protocol uses Pedersen Commitment to hide transaction amounts for confidential transactions, and uses a range proof to prevent users from launching spill attacks. In [12], Monero not only uses a one-way accumulator and knowledge signature, but also uses Pedersen Commitment to hide the transaction amount. In zerocoin [7], Pedersen Commitment is used to generate commitment for newly generated coins. In zerocash [8], the transaction amount is hidden in a commitment. In 2017, Sun et al. [12] proposed the RingCT 2.0 protocol, in which the function of ring signature (linkable ring signature) is implemented by a one-way accumulator and a knowledge signature. Compared to RingCT protocol, this protocol is no longer at the expense of the size of the transaction.

Coin mixing is a technology to enhance anonymity by mixing the coins of multiple users. In 2013, Maxwell [13] proposed Coinjoin; in this protocol, many users can mix their inputs together; through the mix, the user’s input and output are not linkable. In 2014, Mixcoin protocol was proposed by Bonneau et al. [14]; in this protocol, the user sends the coin to a reliable coin-mixing server for coin mixing, and then the server returns the mixed coin to the user. Although the protocol guarantees the unlinkability of the input and output addresses, it introduces a trusted server that knows the information about a user’s input and output addresses, so the server may reveal private information and be prone to single points of failure. This year, Moreno-Sanchez et al. [15] proposed CoinShuffle to enhance the anonymity of Bitcoin, CoinShuffle protocol is a fully decentralized coin mixing protocol that does not require a trusted third party. In the protocol, the user mixes his own coin with the coins of other people participating in the coin mixing, and the protocol outputs the coin after mixing to a newly created specific address. The protocol also cuts off linkability between input and output addresses. Compared to the Mixcoin protocol, CoinShuffle protocol does not require a trusted center and there is no mixing fee. Dash [16] also uses coin mixing techniques to achieve anonymity. Dash designed a two-tier network. The basic functions of the blockchain (e.g., mining, consensus, etc.) are implemented in the first layer of the network, and the master node performs coin mixing operations on the second layer of the network.

From the related works mentioned above, we can see that since the blockchain was proposed, not only the blockchain technology itself and digital currency have been extensively studied by researchers, but also its privacy protection technology has been a notable research topic, however, at present, there are few studies on privacy protection of account balances.

With the emergence of blockchain technology, blockchain has been widely used in IoT [17], healthcare [18], finance and other fields; the application of blockchain to the financial field can tackle the pain points of costy and complicated business processes in traditional financial industry, so that empowering the inclusive finance area as well as changing the landscape of the industry. Currently, blockchain is widely used in digital currency, payment settlement, digital bills, credit management, equity certificate, stock exchanging, insurance management, and other fields.

In the traditional credit system, third-party credit reporting agencies (such as Public Credit Registries (PCR)) used to conduct credit investigations on borrowers [19,20], but they were vulnerable to privacy leakage and information tampering. When sharing information, banks may suffer from single points of failure. In order to avoid many problems, Chang et al. [21] proposed a business integrity modeling and analysis framework to provide financial, operational and liquidity risk analysis. In recent years, people have begun to use blockchain to solve problems in the banking industry. In 2017, Sun et al. [22] proposed MBDC, a model based on permission blockchain technology; this model improves the scalability of the model and the speed of payment by combining the chain structure and ChainID. The user’s identity information and transaction information on the chain can be separated by the user account address protocol. In 2018, Hu et al. [23] proposed a delay-tolerant payment scheme based on the Ethereum blockchain, which aims to solve the problem that users cannot obtain bank services in real time remotely. By deploying user balance smart contracts, banks record users’ balances in fiat and digital currencies and distribute mining rewards. After successfully establishing a connection with a remote account, the bank can synchronize the user’s balance with other nodes and process the currency exchange request. However, this solution seems to put the user’s balance information and other private information at an unknown risk. In the same year, Wang et al. [24] proposed a theoretical credit model based on blockchain technology, which enables small and medium-sized enterprises to evaluate bank loans by using blockchain technology; it also achieves a distributed consensus record of success of debt repayment or debt default rendered by using blockchain technology. In the same year, Goharshady et al. [25] proposed a scheme that uses a blockchain smart contract to generate a credit report required by a bank loan. In this scheme, the credit report does not need to be generated by a credit reporting center, which can solve the problems of slow information update, inconsistent data, unavailability of information, and leakage of sensitive customer information of traditional credit reporting center. The scheme uses cryptographic technology to protect sensitive information that appears in the report. With this scheme, banks or customers can verify whether the report contains the required information that within a certain period of time. In 2018, Godfrey-Welch et al. [26] applied private blockchain technology to payment card transactions. Private blockchain uses public key cryptology to verify cardholder identity and records transactions; it can provide necessary verification information without introducing third parties by using private blockchain technology. In 2019, Wang et al. [27] proposed a new data privacy management framework which is used for financial sector, this framework is based on blockchain technology. The framework is composed of a data privacy classification method, a collaborative-filtering-based model, and a data disclosure confirmation scheme for customer strategies. The framework is used for opening banks. In 2019, Yang et al. [28] proposed a new loan system based on smart contract. To resolve the problem of the traditional loan system, the new loan system combines blockchain and smart contract, it can improve regulatory capacity and loan efficiency. In 2019, Mohamed et al. [29] proposed an approach based on blockchain; it can be used to track money by the serial numbers.

In addition to many experts and scholars beginning to explore the application of blockchain in banks, banks themselves are also actively looking for the combination of blockchain and banks’ business, but it can also be seen that there is not much research on bank loans at present. This article intends to combine blockchain technology and cryptographic technology to design a over-loan prevention mechanism that can protect the data privacy of borrowers and banks. At present, the credit report is mainly generated by a credit reporting agency. The credit report mainly contains the following information: identity information, which include basic information such as name, ID number, and contact information, borrower’s previous loan information and lease in the borrower’s name, details of the borrower’s credit report, utilities, medical expenses, et al., as well as some other public information [30]. It can be found that in the credit report, in addition to some public information of the borrower, there are some private information, such as the borrower’s loan amount in other banks. If these amounts are hidden, the bank may not be able to correctly judge the borrower’s repayment ability, resulting in the problem of excessive loans. Moreover, Goharshady et al. [25] pointed out that the report issued by the credit reporting agency may contain erroneous information. Therefore, this paper makes full use of cryptographic technology to protect the sensitive information and verify whether the loan is an excess loan. It is meaningful to use blockchain to enhance bank participation, reduce the role of credit bureaus in loans, and increase data credibility. At the same time, the algorithm used in this article is different from the application of blockchain in the medical field. As in [31], the author uses attribute-based encryption and cross-domain access strategies to enable users of different medical institutions to access patient data. This paper uses the elliptic curve encryption algorithm to encrypt the data, uses the range proof, Pedersen Commitment and other algorithms to achieve public verification of private data, only the bank that handles the business can access the plaintext data, and other banks and audit nodes can verify the validity of the data by accessing the proof data. Although both papers are aimed at data privacy protection for cross-domain scenarios, due to different restrictions on access to data, the algorithms used are different as well.

## 3. Preliminary

In this paper, we use ring signature, Pedersen Commitment, Elliptic Curve Cryptography (ECC) Algorithm, and so on, to protect the borrower’s private information. This section will review the definition of the above algorithm.

### 3.1. Ring Signature

In our mechanism, we use a variant of Borromean Ring Signature (BRS) proposed in [32] to generate the ring signature of the maximum/loan/repayment amount. The bank and the audit node can verify whether the amount is positive and whether the amount is within a certain range by combining the signature and the range proof, that is, whether the borrower is over-loan.

**Definition** **1.**
*An algorithm that uses its own signature private key and group member public key as signature keys. The algorithm does not require the consent of other members when signing, and even other members do not know that their public key is one of the signature keys. The algorithm has unconditional anonymity and unforgeability.*


Let $\prod_{ringsig}=\{\mathrm{KeyGen},\mathrm{RSig},\mathrm{RVer}$} be a ring signature algorithm. In this algorithm, KeyGen generates the signature keys sk and verification keys pk that based on an elliptic curve, the order of the elliptic curve is q, the base point is G, and another point selected from elliptic curve is H whose discrete relationship with G is unknown, q and p are two large prime numbers that p | q-1. RSig generates a signature π when inputting signature keys and message m. RVer inputs a tuple (π, m, pk) and outputs 1 if the signature π is valid, otherwise 0.

### 3.2. Range Proof

**Definition** **2.**
*Range proof can verify that the number is in a correct range when the number is hidden, because only unsigned integers can pass the range proof verification.*


The ring signature public key used in this scheme is generated based on the value of the jth bit in the binary representation of the loan/payment amount. When the jth bit value (i.e., $a_{j}$) is 0, the ring signature public keys are ${\mathrm{CM}}_{0}$ = $a_{j}2^{j}G$ + $r_{j}H$ = $r_{j}H$ and ${\mathrm{CM}}_{1}$ = ${\mathrm{CM}}_{0}$ − $2^{j}G$ = $r_{j}H$ − $2^{j}G$; When the jth bit value (i.e., $a_{j}$) is 1, the ring signature public keys are ${\mathrm{CM}}_{0}$ = $a_{j}2^{j}G$ + $r_{j}H$ and ${\mathrm{CM}}_{1}$ = $r_{j}H$. Therefore, no matter the value of the jth bit is 0 or 1, the borrower can use $r_{j}$ as the private key and $r_{j}H$ as the verification public key to generate a valid ring signature. When the value of jth bit is neither 0 nor 1 the borrower cannot generate a valid ring signature with $r_{j}$ as the private key and $r_{j}H$ as the verification public key, the bank or audit node cannot prove the validity of the ring signature during verification, and the verification will not pass. Therefore, the success of the signature verification can prove that the value of the jth bit is 0 or 1. By verifying l ring signatures, it can be proved that the borrower’s amount is within the range [0, $2^{l}$).

### 3.3. Elliptic Curve Cryptography (ECC) Algorithm

In the proposed mechanism, during the loan or repayment process, the borrower needs to send his loan/repayment amount to the bank that handles the loan/repayment business. In order to achieve the secure transmission of data, even if an attacker obtains the request sent to the bank by the borrower, he/she cannot obtain the hidden data, and the borrower uses the Elliptic Curve Cryptography (ECC) algorithm [33] to encrypt the amount to ensure secure transmission.

**Definition** **3.**
*The algorithm based on an elliptic curve converts the input plaintext into ciphertext (i.e., two points on the elliptic curve), and without the decryption key, the ciphertext will not be readable by others. The ECC algorithm is based on the elliptic curve discrete logarithm problem.*


Let $\prod_{ecc}$ = {KeyGen, Enc, Dec} be an elliptic curve cryptography algorithm. In this algorithm, KeyGen generates a key pair (sk, pk), where sk is secret key, pk is public key. Enc is used to obtain ciphertext ${\mathrm{enc}}_{m}$ when inputting the tuple (m, pk). Dec decrypts the ${\mathrm{enc}}_{m}$ with sk to obtain the message m. Of course, the correct message can be decrypted only if the private key is correct. If the private key is incorrect, the plaintext cannot be obtained.

### 3.4. Pedersen Commitment

The Pedersen commitment [4] is used to commit a amount, which can be the loan amount, or the repayment amount. During the loan process, because all data are hidden, if the borrower wants to over-loan, he/she may modify the hidden data so that the data recorded by the bank is less than the actual amount of the borrower’s loan. Because the commitment is binding, after the commitment of the amount is generated, if the borrower modifies the loan/repayment amount presented to the bank without modifying the corresponding commitment, the bank will not pass the verification; in addition, the commitment is hidden and if the borrower or bank does not disclose its loan/repayment amount, no one else obtains the loan/repayment amount of the borrower through commitment.

**Definition** **4.**
*The Pedersen Commitment we used is based on the elliptic curve discrete logarithm problem. it consists of two phase, the commit phase and the open phase. In this protocol, Alice commits a value to Bob, but the value is temporarily hidden and Bob does not know what the value is. After a period of time, Alice sends the value hidden in the commitment to Bob, and Bob can verify that the value has not been modified before it was broadcast by Alice. Hiding and binding are two properties of the Pedersen commitment.*


Let $\prod_{commit}$ = {KeyGen, Commit, Open} be a Pedersen commitment. In this protocol, KeyGen selectes some parameters, the lager prime number p, the generator g of $Z_{p}^{*}$ and a random number y belonging to $Z_{p}^{*}$. Commit outputs a commitment when inputting the value that needs to be committed, the generator g, the random number y and another random number r belonging to $Z_{p}^{*}$. Open publishes the committed value and the random number r, and outputs 1 if the commitment has not been modified, or 0 otherwise.

### 3.5. Elliptic Curve Digital Signature Algorithm (ECDSA)

In this mechanism, in order to verify that the commitment of the amount is honestly generated by the borrower, it is necessary to prove that the commitment $C_{r}$ = rH of the random number r is honestly generated by the borrower, that is, the commitment $C_{r}$ is only related to point H, not to point G. Therefore, it is necessary to generate a proof about the discrete relationship between point H and the commitment $C_{r}$. Through the proof, it can prove that the borrower did not hide part of the amount in the commitment $C_{r}$ to achieve the purpose of over-loan. In the ECDSA [34], the borrower uses the random number r as the private key and $C_{r}$ as the public key to generate a signature. By judging the validity of the signature, the bank judges whether $C_{r}$ is only related to point H.

**Definition** **5.**
*The ECDSA is based on the elliptic curve discrete logarithm problem. In this algorithm, the singer can sign a message m by a private key, and the verifier can verify the signature by a public key. This algorithm is non-repudiation, integrity and authentication.*


Let $\prod_{ECDSA}$ = {KeyGen, Sign, Verify} be an elliptic curve digital dignature algorithm. KeyGen generates a key pair (sk, pk), where sk is secret key, pk is public key. The curve used in KeyGen is an elliptic curve over a finite field GF(p), the elliptic curve’s base point is G, and the order of G is n. Sign outputs a message m’s signature π when inputting the message m, sk and a random number k selected by signer. When the tuple (π, m, pk) is input, Verify outputs 1 if the signature π is valid, or 0 otherwise.

### 3.6. Bulletproofs Algorithm

In the CBOL-bullet, we use Bulletproofs algorithm proposed in [35] to reduce the size of range proof. The Bulletproofs algorithm does not require trusted settings and only depends on the discrete logarithm problem. The range proof size generated by the algorithm is logarithmic. The Bulletproofs algorithm can verify a secret value is within a given range, and the inner product is used to reduce the size of the range proof.

Let $\prod_{bullet}$ = {KeyGen, GenRange, VerRange} be a bulletproofs algorithm. KeyGen takes the safety parameter λ as input and takes H, G, g, h as output, where H and G are two points on the elliptic curve, and the discrete relationship is unknown, and g and h are the elliptic curve dots vector. GenRange generates the range proof of the amount. The prover first converts the amount into a binary string $a_{L}$ =${\left\{0,1\right\}}^{l}$, computes $a_{R}$ =$\left\{a_{L_{1}}-1,\ldots ,a_{L_{l}}-1\right\}$, then chooses the random numbers $r_{1},r_{2}\in Z_{q}$ and $s_{L},s_{R}\in Z_{q}^{l}$, computes $A=H^{r_{1}}g^{a_{L}}h^{a_{R}}$, $S=H^{r_{2}}g^{s_{L}}h^{s_{R}}$. After receiving the random numbers y and z sent by the verifier, the prover constructs polynomial t(X)=<l(X),r(X)>, chooses the random numbers $\tau_{1}$ and $\tau_{2}$, and then computes $T_{1}=G^{t_{1}}H^{\tau_{1}}$ and $T_{2}=G^{t_{2}}H^{\tau_{2}}$, where $\tau_{1}$ and $\tau_{2}$ are the coefficient of the polynomial t(X). After receiving the random number x sent by the verifier, the prover generates $\tau_{x},\mu ,t\left(x\right),l(x),r(x),h^{\prime },h^{\prime }$ and $P^{\prime }$. Finally, the prover sends the range proof generated to the verifier. After receiving the range proof from the verifier, VerRange verifies whether the range proof is valid. Firstly, the verifier computes $h^{\prime }$, and judges whether t(x) is equal to <l(x),r(x)>, then the verifier computes $G^{t}H^{\tau_{x}}$ and P, judges whether P is equal to $H^{\tau }g^{l(x)}h^{\prime }r(x)$. If the verification is passed, the amount hidden in the range proof is in the range [0, $2^{l}$− 1]; otherwise, it indicates that the amount is overflow.

## 4. System Structure and Adversarial Model

In this section, we will discuss the system structure and adversarial model. Identity management enables the system to be executed in an organized and orderly manner. Literature [36] is to realize the dynamic deployment of SFC through the connection between the different roles, so as to minimize the costs under the premise of meeting the conditions. In this paper, a goal is achieved by assigning different functions to different roles. Those roles can be connected through p2p network.

### 4.1. System Structure

The mechanisms proposed in this paper mainly includes four parts: borrowers, banks, audit nodes and blockchain. The functions of each character are as follows:
**Borrower**. The borrower loans from or repays a bank which is a bank union. During the loan/repayment process, the borrower generates the ciphertext of the loan/repayment amount, the commitment of the loan/repayment amount, the commitment of the new remaining loanable amount, the range proof of the loan/repayment amount, the range proof of the new remaining loanable amount, and the knowledge proof of the corresponding random number. The borrower then sends all the data to a bank. The borrower can communicate with all banks and perform related calculations offline.**Bank**. All banks can use this mechanism to prevent borrowers from over-loaning form a bank union. The bank union needs to evaluate a borrower’s repayment ability and set the corresponding maximum loan limit. Each bank needs to generate a public/private key pair for the borrower to securely transfer the loan/repayment amount. The bank dealing with the loan/repayment also needs to verify the validity of the proof data related to the loan/repayment amount, that is, to determine whether the borrower modifies the amount and whether the borrower is over-loan. If the data are valid, the bank sends the data to the blockchain network and agrees to the borrower’s loan/repayment request; otherwise, the bank rejects the borrower’s loan/repayment request, and does not post the data to the blockchain network. Banks can communicate with all audit nodes and can access to data saved on the blockchain.**Audit node**. The audit node can be the committers in the blockchain. The audit node can verify the validation of the transaction issued by the bank node in the blockchain network, that is, whether the range proof of the amount meets the requirements (the amount is positive), and whether the borrower is over-loan. The audit node updates the relevant proof of the amount on the chain, if the data are valid; otherwise, the audit node rejects the transaction. The audit nodes communicate through the P2P network.**Blockchain**. As a distributed ledger, the blockchain is responsible for storing valid data.

### 4.2. Adversarial Model

In this section, we define two types of attacks; one is an attack on random numbers, and another is an attack on amounts.
**Attack on random numbers**. In this attack, when the borrower is the attacker, the borrower generates a random number to achieve the purpose of over loan; when the bank or audit node is the attacker, they want to obtain the random number by cracking the commitment of random number.**Attack on amount**. In this attack, when the borrower is an attacker during a loan or repayment process, the borrower modifies the amount hidden in the commitment to achieve the purpose of over loan; when the bank or audit node is an attacker, they obtain the loan/repayment amount of the borrower in a certain bank by cracking the ciphertext or the commitment.

## 5. Blockchain-Based Across-Bank Over-Loan Prevention Mechanism

In this section, we introduce CBOL-ring mechanism we designed in details. The mechanisms are divided into four stages: register, initialization, loan, and repayment. In the following, we will introduce the behavior of each role at different stages. Finally, we theoretically analyzed the range proof size of the proposed mechanism. The Figure 1 shows framework of the mechanism we proposed.

### 5.1. CBOL-Ring Mechanism

**Stage 1: registration.** In this stage, a borrower sends a *registration request* to a bank union, and the bank union returns some data to the borrower. The detailed description is as follows:
Borrower: When a borrower wants to register, he/she sends a *registration request* to the bank union.Bank Union: Once the bank union receives a *registration request* from a borrower, it
–sets a identity number ID for the client.–sets a maximum loan amount $a_{\max}$.–generates n public/private key pairs (${\mathrm{pk}}_{i},{\mathrm{sk}}_{i}$), where i∈[1,n], n is the number of banks in the bank alliance.–sends ID, $a_{\max}$, ${\mathrm{pk}}_{1}$,..., ${\mathrm{pk}}_{n}$ to the client.

**Stage 2: Initialization.** This stage is mainly used to generate proof data of the maximum loan limit $a_{\max}$ and store them on the blockchain, so that banks and audit nodes can verify the validity of the loan or repayment stage data through the relevant proof data of the maximum loan limit $a_{\max}$. In this stage, the borrower sends a *initialization request* to the bank union. Once the bank union receives the *initialization request*, it verifies whether the data contained in the request is valid, if the data are valid, the bank union sends all the data to the audit node, otherwise, the bank union rejects the request. When the audit nodes receive all the data, they verify whether all the data are valid; if the data are valid, the data are uploaded to blockchain, otherwise, they reject the data. Figure 2 shows the process of the initialization stage. The detailed process is as follows:
Borrower: When the borrower wants to register in the banking union, he/she
–computes the maximum loan amount $a_{\max}$’s commitment $C_{\max}$ and the random number $r_{\max}$’s commitment $C_{r_{\max}}$, the equation is as follows:
(1)$$C_{max}=a_{max}\cdot G+r_{max}\cdot H.$$
(2)$$C_{r_{max}}=r_{max}\cdot H.$$
where $r_{\max}$ is a random number corresponding to $a_{\max}$.–generates the knowledge proof $\sigma_{r_{\max}}$ about discrete relationship between $C_{r_{\max}}$ and the point H by signing the massage m with the private key $r_{\max}$, the equation is as follows:
(3)$$\sigma_{r_{max}}=Sign\left(r_{max},m,C_{r_{max}}\right).$$–sends ID, $a_{\max}$, m, $C_{r_{\max}}$, $C_{\max}$, $\sigma_{r_{\max}}$ to the bank union.Bank Union: When the bank union receives the *initialization request*, any bank
–computes commitments with the maximum loan amount $a_{\max}$ received, the equation is as follows:
(4)$$C_{max}^{\prime }=a_{max}\cdot G+C_{r_{max}}.$$–judges whether $C_{max}^{\prime }$ is equal to *C_max_*.–verifies whether the signature $\sigma_{r_{\max}}$ is legal.–sends ID, $a_{\max}$, m, $C_{r_{\max}}$, $C_{\max}$, $\sigma_{r_{\max}}$ to audit node, if all the verification is valid; otherwise, the bank union rejects the *initialization request*.Audit Nodes: When the audit nodes receive all the data from the bank union, they also
–compute the commitment $C_{max}^{\prime \prime }$ with the maximum loan amount $a_{\max}$ received, the equation is as follows:
(5)$$C_{max}^{\prime \prime }=a_{max}\cdot G+C_{r_{max}}.$$–judge whether $C_{max}^{\prime \prime }$ is equal to *C_max_*.–verify whether the signature $\sigma_{r_{\max}}$ is legal.
(6)$$Ver(\sigma_{r_{max}},m,C_{r_{max}})?=1.$$–upload $T_{\mathrm{init}}$ = {ID, $a_{\max}$, $C_{r_{\max}}$, $C_{\max}$, $\sigma_{r_{\max}}$} to blockchain, if all the verification is valid; otherwise, the audit nodes reject the $T_{\mathrm{init}}$.

**Stage 3: Loan** In this stage, the borrower wants to loan $a_{\mathrm{loan}}$ from a bank (for example bank 1) in the bank union, the borrower sends a *loan request*, and the *loan request* contains the corresponding data. Then the bank verifies all the data, if the data are valid, it sends some data to the audit nodes; otherwise, it rejects the request. Once the audit nodes receive the data, it also verify the verification of the data to decide whether to upload the data to the blockchain or reject the data. Figure 3 shows the process of the loan stage.
Borrower: It assumes that the borrower’s current remaining loanable amount is $a_{\mathrm{old}}$, the current remaining loanable amount commitment is $C_{\mathrm{old}}$, the random number for generating the commitment $C_{\mathrm{old}}$ is $r_{\mathrm{old}}$, that is $C_{\mathrm{old}}$ = $a_{\mathrm{old}}G$ + $r_{\mathrm{old}}H$, and the borrower’s current loan amount is $a_{\mathrm{loan}}$, the remaining loanable amount after this loan is $a_{\mathrm{new}}$, and satisfies $a_{\mathrm{new}}$ = $a_{\mathrm{old}}$−$a_{\mathrm{loan}}$, pk is the public key of the bank handling the loan. Once the borrower decides to loan from the bank, he/she
–computes the commitments, the equation is as follows:
(7)$$C_{loan}=a_{loan}\cdot G+r_{loan}\cdot H.$$
(8)$$C_{new}=a_{new}\cdot G+r_{new}\cdot H.$$
(9)$$C_{r_{loan}}=r_{loan}\cdot H.$$
where $C_{\mathrm{loan}}$ is a commitment of the loan amount, $C_{\mathrm{new}}$ is a commitment of the new remaining loan amount and $C_{r_{\mathrm{loan}}}$ is a commitment of the random number.–generates the knowledge proof $\sigma_{r_{loan}}$ about the discrete relationship between $C_{r_{\mathrm{loan}}}$ and point H by signing the massage m with the private key $r_{\mathrm{loan}}$, the equation is as follows:
(10)$$\sigma_{r_{loan}}=Sign\left(r_{loan},C_{r_{loan}},m\right).$$–generates the ring signatures of $a_{\mathrm{loan}}$ and $a_{\mathrm{new}}$, the equation is as follows:
(11)$$\pi_{a_{loan}}=RSign\left(sk_{loan},pk_{1,loan},pk_{2,loan}\right).$$
(12)$$\pi_{a_{new}}=RSign\left(sk_{new},pk_{1,new},pk_{2,new}\right).$$
where ${\mathrm{sk}}_{\mathrm{loan}}$ and ${\mathrm{sk}}_{\mathrm{new}}$ are the private keys of the signers in the ring signature, (${\mathrm{pk}}_{1,\mathrm{loan}}$, ${\mathrm{pk}}_{2,\mathrm{loan}}$) and (${\mathrm{pk}}_{1,\mathrm{new}}$, ${\mathrm{pk}}_{2,\mathrm{new}}$) are the two public keys of the ring signature respectively.–encrypts the loan amount $a_{\mathrm{loan}}$ using the ECC algorithm to obtain the ciphertext ${\mathrm{enc}}_{a_{\mathrm{loan}}}$, the equation is as follows:
(13)$$enc_{a_{loan}}=Enc\left(a_{loan},pk\right).$$sends ID, m, $C_{\mathrm{loan}}$, $C_{\mathrm{new}}$, ${\mathrm{enc}}_{a_{\mathrm{loan}}}$, $C_{r_{\mathrm{loan}}}$, $\pi_{a_{\mathrm{loan}}}$, $\pi_{a_{\mathrm{new}}}$, $\sigma_{r_{\mathrm{loan}}}$ to the bank.Bank 1: When the bank receives the data from the borrower, it
–decrypts the ciphertext ${\mathrm{enc}}_{a_{\mathrm{loan}}}$ to obtain the borrower’s loan amount.
(14)$$a_{loan}^{\prime }=Dec\left(enc_{a_{loan}},sk\right).$$
where sk is the bank’s private key.–computes the commitment $C_{\mathrm{loan}}^{\prime }$ with the loan amount $a_{\mathrm{loan}}^{\prime }$ received from the borrower.
(15)$$C_{loan}^{\prime }=a_{loan}^{\prime }\cdot G+pk_{sign_{loan}}.$$–judges whether $C_{\mathrm{loan}}^{\prime }$ is equal to $C_{\mathrm{loan}}$.–verifies the relationship between $C_{\mathrm{loan}}$, $C_{\mathrm{new}}$ and $C_{\mathrm{old}}$, where $C_{\mathrm{old}}$ is the commitment of the remaining loan amount after the last loan/repayment, and it can be obtained from blockchain. The correct relationship between the three should be
(16)$$C_{old}=C_{loan}+C_{new}.$$–verifies that the three signatures $\pi_{a_{\mathrm{loan}}}$, $\pi_{a_{\mathrm{new}}}$ and $\sigma_{\mathrm{loan}}$ are valid.
(17)$$RVer(\pi_{a_{loan}},pk_{1,loan},pk_{2,loan})?=1.$$
(18)$$RVer(\pi_{a_{new}},pk_{1,new},pk_{2,new})?=1.$$
(19)$$Ver(\sigma_{r_{loan}},m,C_{r_{loan}})?=1.$$–sends ID, $C_{\mathrm{loan}}$, $C_{\mathrm{new}}$, $\pi_{a_{\mathrm{loan}}}$, $\pi_{a_{\mathrm{new}}}$ to the audit nodes, if all the verifications are correct; otherwise, the bank rejects the *loan request*.Audit Nodes: When the audit nodes receive all the data from the bank, they also
–verify that the two ring signatures $\pi_{\mathrm{loan}}$. $\pi_{\mathrm{new}}$ are valid.
(20)$$RVer(\pi_{a_{loan}},pk_{1,loan},pk_{2,loan})?=1.$$
(21)$$RVer(\pi_{a_{new}},pk_{1,new},pk_{2,new})?=1.$$–upload $T_{\mathrm{loan}}$ = {ID, $C_{\mathrm{loan}}$, $C_{\mathrm{new}}$, $\pi_{a_{\mathrm{loan}}}$, $\pi_{a_{\mathrm{new}}}$} to blockchain.

**Stage 4: Repayment.** In this phase, the borrower can repay his/her loan, the process is similar to the loan process. When the borrower wants to repay the loan, he/she sends a *repayment request* to the bank which he/she loan before. Once the bank receives the data contained in the request, it verifies the verification of the data. If all the data are valid, the bank sends some of the data to the audit nodes; otherwise, the bank rejects the *repayment request*. When the audit nodes receive the data from the bank, they also verify the verification of the data, if the data are valid, the audit nodes upload the data to blockchain; otherwise, they reject the data. Figure 4 shows the process of the repayment stage.
Borrower: It assumes that the borrower’s current remaining loanable amount is $a_{\mathrm{old}}^{{}^{\prime }}$, the current remaining loanable amount commitment is $C_{\mathrm{old}}^{{}^{\prime }}$, the random number for generating the commitment $C_{\mathrm{old}}^{{}^{\prime }}$ is $r_{\mathrm{old}}^{{}^{\prime }}$, that is $C_{\mathrm{old}}^{{}^{\prime }}$ = $a_{\mathrm{old}}^{{}^{\prime }}G$ + $r_{\mathrm{old}}^{{}^{\prime }}H$, and the borrower’s current repayment amount is $a_{\mathrm{repay}}$, the remaining loanable amount after this repayment is $a_{\mathrm{new}}^{{}^{\prime }}$, and satisfies $a_{\mathrm{new}}^{{}^{\prime }}$ = $a_{\mathrm{old}}^{{}^{\prime }}$ + $a_{\mathrm{repya}}$, ${\mathrm{pk}}^{{}^{\prime }}$ is the public key of the bank handling the repayment. When the borrower wants to repay his/her loan, he/she sends a *repayment request*. He/She
–computes the commitments:
(22)$$C_{repay}=a_{repay}\cdot G+r_{repay}\cdot H.$$
(23)$$C_{new}^{\prime }=a_{new}^{\prime }\cdot G+r_{new}^{\prime }\cdot H.$$
(24)$$C_{r_{repay}}=r_{repay}\cdot H.$$–generates the ring signatures of $a_{\mathrm{repay}}$ and $a_{\mathrm{new}}^{\prime }$, the equation is as follows:
(25)$$\pi_{a_{repay}}=RSign\left(sk_{repay},pk_{1,repay},pk_{2,repay}\right).$$
(26)$$\pi_{a_{new}^{\prime }}=RSign\left(sk_{new}^{\prime },pk_{1,new}^{\prime },pk_{2,new}^{\prime }\right).$$–generates the knowledge proof $\sigma_{\mathrm{repay}}$ about the discrete relation between $C_{r_{\mathrm{repay}}}$ and the point H by signing the massage m with the private key $r_{\mathrm{repay}}$, the equation is as follows:
(27)$$\sigma_{r_{repay}}=Sign\left(r_{repay},C_{r_{repay}},m\right).$$–encrypts the repayment amount $a_{\mathrm{repay}}$ using the ECC algorithm to protect the amount during the transfer, the equation is as follows:
(28)$$enc_{a_{repay}}=Enc\left(a_{loan},pk^{\prime }\right).$$–sends ID, m, ${\mathrm{enc}}_{a_{\mathrm{repay}}}$, $C_{\mathrm{repay}}$, $C_{\mathrm{new}}^{\prime }$, $C_{r_{\mathrm{repay}}}$, $\pi_{a_{\mathrm{repay}}}$, $\pi_{a_{\mathrm{new}}^{\prime }}$, $\sigma_{r_{\mathrm{repay}}}$ to the bank.Bank 1: After receiving the data, the bank
decrypts ${\mathrm{enc}}_{a_{\mathrm{repay}}}$ to obtain the repayment amount $a_{\mathrm{repay}}^{\prime }$, the equation is as follows:
(29)$$a_{repay}^{\prime }=Dec\left(enc_{a_{repay}},sk^{\prime }\right).$$
where ${\mathrm{sk}}^{\prime }$ is the bank’s private key.–computes the commitment $C_{\mathrm{repay}}^{\prime }$ of $a_{\mathrm{repay}}^{\prime }$ to determine whether the repayment amount $a_{\mathrm{repay}}$ has been modified, the equation is as follows:
(30)$$C_{repay}^{\prime }=a_{repay}^{\prime }\cdot G+C_{r_{repay}}.$$–judges whether $C_{\mathrm{repay}}$ is equal to $C_{\mathrm{repay}}^{\prime }$.–verifies the relationship between $C_{\mathrm{repay}}$, $C_{\mathrm{new}}^{\prime }$ and $C_{\mathrm{old}}^{\prime }$, where $C_{\mathrm{old}}^{\prime }$ is the commitment of the remaining loan amount after the last loan/repayment, and it can be obtained from blockchain. The correct relationship between the three should be:
(31)$$C_{new}^{\prime }=C_{old}^{\prime }+C_{repay}.$$–verifies that the three signatures $\pi_{a_{\mathrm{repay}}}$, $\pi_{a_{\mathrm{new}}^{\prime }}$ and $\sigma_{a_{\mathrm{repay}}}$ are valid.
(32)$$RVer(\pi_{a_{repay}},pk_{1,repay},pk_{2,repay})?=1.$$
(33)$$RVer(\pi_{a_{new}^{\prime }},pk_{1,new}^{\prime },pk_{2,new}^{\prime })?=1.$$
(34)$$Ver(\sigma_{r_{repay}},m,C_{r_{repay}})?=1.$$–sends ID, $C_{\mathrm{repay}}$, $C_{\mathrm{new}}^{\prime }$, $\pi_{a_{\mathrm{repay}}}$, $\pi_{a_{\mathrm{new}}^{\prime }}$ to the audit nodes, if all the data are verification; otherwise, if one of the data are fails to pass, the bank ends the verification process, and rejects the *repayment request* from the borrower.Audit node: When the audit nodes receive the data from the bank, they also
–verify that the two ring signature $\pi_{a_{\mathrm{repay}}}$, $\pi_{a_{\mathrm{new}}^{\prime }}$ are valid.
(35)$$RVer(\pi_{a_{repay}},pk_{1,repay},pk_{2,repay})?=1.$$
(36)$$RVer(\pi_{a_{new}^{\prime }},pk_{1,new}^{\prime },pk_{2,new}^{\prime })?=1.$$–uploan $T_{\mathrm{repay}}$ = (ID, $C_{\mathrm{repay}}$, $C_{\mathrm{new}}^{\prime }$, $\pi_{a_{\mathrm{repay}}}$, $\pi_{a_{\mathrm{new}}^{\prime }}$) to blockchain; otherwise, if one of the signatures is not valid, the audit nodes reject the data from the bank.

### 5.2. Size of Range Proof

In the CBOL-ring mechanism, only the size of range proofs is affected by the length of the binary string, so in this section, we theoretically analyze the relationship between the range proof size and the length of the binary string.

It can be found from Table 1 that when the length of the binary string is l, the range proof generated contains l elliptic curve elements and 5l finite field elements, that is, the range proof size is linear with the length of the binary string.

## 6. Blockchain-Based Cross-Bank Over-Loan Prevention Mechanism with Low Communication Volume

In the CBOL-ring mechanism proposed in Section 5, a combination of ring signature algorithm and range proof can verify that a value is in a specific range. However, through analysis, it is found that when the amount is divided into bits, the communication volume is affected by the length of the binary string, and is basically linear with the length of the binary string. In the CBOL-ring mechanism, the borrower can choose the length of the binary string that meets the basic security when generating the range proof, but if higher security is required, the length of the string must be increased. At this time, the CBOL-ring mechanism will cause communication volume problems. When the communication volume is large, it will not only affect the communication efficiency, but also affect the probability of a successful transaction into the block, because each block in the blockchain has a size limit, when the storage space required for a transaction is too large, even the transaction is legal, the transaction may not be able to enter the blockchain normally. Therefore, in order to meet the requirements of different degrees, it is necessary to reduce the impact of the length of the binary string on the communication volume by using an efficient range proof algorithm.

The CBOL-bullet uses the Bulletproofs algorithm to change the way in which range proofs are generated to reduce the amount of communication during the loan and repayment stage. In the optimized mechanism, the way to generate the commitment of the amount, the commitment of random number, the proof about the discrete relation between $C_{r_{\mathrm{repay}}}$ and the point H, and the ciphertext of amount are all consistent with the CBOL-ring mechanism. The following introduces the CBOL-bullet.

### CBOL-Bullet

**Stage 1: Registration.** All banks in the banking union generate ID for borrowers, the maximum loan limit $a_{\max}$, and the public and private keys (${\mathrm{pk}}_{i}$, ${\mathrm{sk}}_{i}$) used by the ECC algorithm, and then the ID, $a_{\max}$ and all public keys ${\mathrm{pk}}_{i}$ are sent to the borrower, where i = 1, 2, …, k, k is the number of banks in the banking union.

**Stage 2: Initialization.** This stage is the same as the initialization stage in the CBOL-ring scheme.
Borrower: After registering, the borrower needs to generate the proof for the maximum loan limit $a_{\max}$.
–generates the commitment $C_{\max}$ of $a_{\max}$ and the commitment $C_{r_{\max}}$ of random numbers $r_{\max}$, as well as the knowledge proof about the discrete relationship between $C_{r_{\max}}$ and the point H.–sends ID, $a_{\max}$, m, $C_{r_{\max}}$, $C_{\max}$, $\sigma_{r_{\max}}$ to the bank union.Bank union: When the bank union receives the data form the borrower, any bank in the banking union
–verifies the validity of the commitment $C_{\max}$ and the signature $\sigma_{r_{\max}}$.–sends ID, $a_{\max}$, m, $C_{r_{\max}}$, $C_{\max}$, $\sigma_{r_{\max}}$ to audit node, if all the data are valid; otherwise, the bank union rejects all the data.Audit node: When the audit node receives the data from the bank, they also
–verify the validity of the commitment $C_{\max}$ and the signature $\sigma_{r_{\max}}$.–upload $T_{\mathrm{init}}$ = {ID, $a_{\max}$, $C_{r_{\max}}$, $C_{\max}$, $\sigma_{r_{\max}}$} to blockchain, if all the verification is valid; otherwise, the audit nodes reject the $T_{\mathrm{init}}$.

**Stage 3: Loan.** This stage is similar to the loan stage in the CBOL-ring scheme. This stage consists of three parts: the borrower generates the proof, bank verifies the proof, and audit nodes verify the proof.
Borrower: When the borrower wants to loan from a bank, he/she
–calculates the commitment $C_{\mathrm{loan}}$ of the loan amount $a_{\mathrm{loan}}$, the commitment $C_{\mathrm{new}}$ of the new remaining loanable amount $a_{\mathrm{new}}$, and generates the knowledge proof $\sigma_{r_{\mathrm{loan}}}$ about discrete relationship between $C_{r_{\mathrm{loan}}}$ and the point H by the signature function of the ECDSA algorithm, that is, $r_{\mathrm{loan}}$ is used as the private key, and $C_{r_{\mathrm{loan}}}$ is used as the public key to generate the signature, where $a_{\mathrm{new}}$ = $a_{\mathrm{old}}$−$a_{\mathrm{loan}}$, $r_{\mathrm{new}}$ = $r_{\mathrm{old}}$−$r_{\mathrm{loan}}$, $a_{\mathrm{old}}$ and $r_{\mathrm{old}}$ are the borrower’s current remaining loanable amount and the corresponding random numbers.–generates the range proof for the amount by using the GenRange function of the Bulletproofs algorithm. The equation is as follows:
(37)$$\pi_{a_{loan}}=GenRange\left(a_{loan},r_{loan}\right).$$
(38)$$\pi_{a_{new}}=GenRange\left(a_{new},r_{new}\right).$$–encrypts the loan amount $a_{\mathrm{loan}}$ using the ECC algorithm to obtain the ciphertext ${\mathrm{enc}}_{a_{\mathrm{loan}}}$.–sends ID, m, $C_{\mathrm{loan}}$, $C_{\mathrm{new}}$, ${\mathrm{enc}}_{a_{\mathrm{loan}}}$, $C_{r_{\mathrm{loan}}}$, $\pi_{a_{\mathrm{loan}}}$, $\pi_{a_{\mathrm{new}}}$, $\sigma_{r_{\mathrm{loan}}}$ to the bank.Bank: When the bank receives a loan request from the borrower, it
–decrypts the ciphertext of the amount ${\mathrm{enc}}_{a_{\mathrm{loan}}}$ by using the decryption function of the ECC algorithm to obtain the actual loan amount $a_{\mathrm{loan}}^{\prime }$ of the borrower, and calculates the commitment $C_{\mathrm{loan}}^{\prime }$ of the loan amount to verify the validity of the commitment.–verifies the validity of the range proof through the VerRange function of the Bulletproofs algorithm. The equation is as follows:
(39)$$VerRange(\pi_{a_{loan}})?=1.$$
(40)$$VerRange(\pi_{a_{new}})?=1.$$–verifies the validity of the knowledge proof $\sigma_{r_{\mathrm{loan}}}$ by using the validation function of ECDSA algorithm.–sends ID, $C_{\mathrm{loan}}$, $C_{\mathrm{new}}$, $\pi_{a_{\mathrm{loan}}}$, $\pi_{a_{\mathrm{new}}}$ to the audit nodes, if all the verifications are correct; otherwise, the bank rejects all the data.Audit node: When the audit nodes receive the data sent by the bank, they
–verify the validity of the range proof through the VerRange function of the Bulletproofs algorithm. The equation is as follows:
(41)$$VerRange(\pi_{a_{loan}})?=1.$$
(42)$$VerRange(\pi_{a_{new}})?=1.$$–upload $T_{\mathrm{loan}}$ = {ID, $C_{\mathrm{loan}}$, $C_{\mathrm{new}}$, $\pi_{a_{\mathrm{loan}}}$, $\pi_{a_{\mathrm{new}}}$} to blockchain.

**Stage 4: Repayment.** This stage is also similar to the repayment stage in the CBOL-ring mechanism.
Borrower: When the borrower wants to repay, he/she
–calculates the commitment $C_{\mathrm{repay}}$ of the loan amount $a_{\mathrm{repay}}$, the commitment $C_{\mathrm{new}}^{\prime }$ of the new remaining loanable amount $a_{\mathrm{new}}^{\prime }$, and generates the knowledge proof $\sigma_{r_{\mathrm{repay}}}$ about discrete relationship between $C_{r_{\mathrm{repay}}}$ and the point H by the signature function of the ECDSA algorithm, that is, $r_{\mathrm{repay}}$ is used as the private key, and $C_{r_{\mathrm{repay}}}$ is used as the public key to generate the signature, where $a_{\mathrm{new}}^{\prime }$ = $a_{\mathrm{old}}^{\prime }$ + $a_{\mathrm{repay}}$, $r_{\mathrm{new}}^{\prime }$ = $r_{\mathrm{old}}^{\prime }$ + $r_{\mathrm{repay}}$, $a_{\mathrm{old}}^{\prime }$ and $r_{\mathrm{old}}^{\prime }$ are the borrower’s current remaining loanable amount and the corresponding random numbers.–generates the range proof for the amount by using the GenRange function of the Bulletproofs algorithm. The equation is as follows:
(43)$$\pi_{a_{repay}}=GenRange\left(a_{repay},r_{repay}\right).$$
(44)$$\pi_{a_{new}^{\prime }}=GenRange\left(a_{new}^{\prime },r_{new}^{\prime }\right).$$–encrypts the repayment amount $a_{\mathrm{repay}}$ using the ECC algorithm to obtain the ciphertext ${\mathrm{enc}}_{a_{\mathrm{repay}}}$.–sends ID, m, $C_{\mathrm{repay}}$, $C_{\mathrm{new}}^{\prime }$, ${\mathrm{enc}}_{a_{\mathrm{repay}}}$, $C_{r_{\mathrm{repay}}}$, $\pi_{a_{\mathrm{repay}}}$, $\pi_{a_{\mathrm{new}}^{\prime }}$, $\sigma_{r_{\mathrm{loan}}}$ to the bank.Bank: When the bank receives a repayment request from the borrower, it
–decrypts the ciphertext of the amount ${\mathrm{enc}}_{a_{\mathrm{repay}}}$ by using the decryption function of the ECC algorithm to obtain the actual loan amount $a_{\mathrm{repay}}^{\prime }$ of the borrower, and calculates the commitment $C_{\mathrm{repay}}^{\prime }$ of the loan amount to verify the validity of the commitment.–verifies the validity of the range proof through the VerRange function of the Bulletproofs algorithm. The equation is as follows:
(45)$$VerRange(\pi_{a_{repay}})?=1.$$
(46)$$VerRange(\pi_{a_{new}^{\prime }})?=1.$$–verifies the validity of the knowledge proof $\sigma_{r_{\mathrm{repay}}}$ by using the validation function of ECDSA algorithm.–sends ID, $C_{\mathrm{repay}}$, $C_{\mathrm{new}}^{\prime }$, $\pi_{a_{\mathrm{repay}}}$, $\pi_{a_{\mathrm{new}}^{\prime }}$ to the audit nodes, if all the verifications are correct; otherwise, the bank rejects all the data.Audit node: When the audit nodes receive the data sent by the bank, they
–verify the validity of the range proof through the VerRange function of the Bulletproofs algorithm. The equation is as follows:
(47)$$VerRange(\pi_{a_{loan}})?=1.$$
(48)$$VerRange(\pi_{a_{new}^{\prime }})?=1.$$–uploan $T_{\mathrm{repay}}$ = (ID, $C_{\mathrm{repay}}$, $C_{\mathrm{new}}^{\prime }$, $\pi_{a_{\mathrm{repay}}}$, $\pi_{a_{\mathrm{new}}^{\prime }}$) to blockchain; otherwise, if one of the signatures is not valid, the audit node reject the data from the bank.

## 7. Security and Performance Analysis

In this section, we discuss the security of the two mechanisms we proposed.

### 7.1. Security Analysis

**Security of random number**. In this mechanism, the random number is secure. Firstly, the borrower must honestly generate the random number, since the data sent by the borrower to the bank contains a signature, which is generated by using the commitment of the random number as the public key and the random number as the private key. If the borrower generates the random number, and hides part of the amount in the commitment of the random number, then he/she does not know the discrete relationship between the commitment of the random number and the point H, and they cannot generate a valid signature. When the bank verifies that the signature is invalid, it rejects the borrower’s request, so the borrower must generate the random number. Secondly, the bank and the audit node cannot obtain the random number because the random number is hidden in the commitment, which is based on the difficulty of the elliptic curve discrete logarithm problem. It is difficult for the bank and the audit node to solve the problem in the case of limited computing power. So as long as the borrower does not disclose the random number, the bank and the audit node cannot obtain the random number.**Security of amount**. In this mechanism, the amount is secure. Firstly, the borrower must honestly send the amount and cannot modify the amount hidden in the commitment, because the bank verifies the commitment generated by the borrower when verifying the validity of the data. When the verification fails, the bank rejects the borrower’s request. In addition, during the loan/repayment process, the borrower needs to generate a range proof of the amount. When the borrower wants to over loan, the bank cannot verify the validity of the range proof and also rejects the borrower’s request. Secondly, the bank and the audit node cannot obtain the loan/repayment amount of the borrower in a certain bank because the borrower hides the amount in the ciphertext and commitment when sending the loan/repayment request to the bank. The encryption algorithm and Pedersen Commitment are based on the difficulty of the elliptic curve discrete logarithm problem, It is difficult for the bank and the audit node to solve the problem in the case of limited computing power. Hence, as long as the borrower does not disclose the amount, the bank and the audit node cannot obtain the amount.

### 7.2. Performance Analysis

Since the CBOL-bullet is an improvement compares to the CBOL-ring mechanism, the only difference between them are range proofs generation. Therefore, this section analyzes the performance of the CBOL-ring mechanism and comparison on performance of the two mechanisms.

#### 7.2.1. Experimental Environment Settings

We deploy the CBOL-ring scheme and the CBOL-bullet in the Ubuntu 18.04 environment of the Acer laptop. The parameters of the laptop are 2.50-GHz, Intel (R) Core (TM), i7-6500U CPU, 8.00GB RAM. The elliptic curve signature algorithm library [37] is used to implement the elliptic curve point multiplication operation and the loan/repayment amount encryption operation, the sha-256 hash algorithm is used as the hash function involved in the scheme, and the GNU Multiple Precision (GMP) algorithm library [38] implements a large integer operation. In order to ensure security, the elliptic curve we used is secp256k1 [39].

In order to fully verify the impact of the change in binary length on the running time, in this experiment, the maximum loan limit of the borrower is set to 70,000, the current loan amount is set to 1, the current remaining loan amount is set to 1, and the length of the binary string is set to 2, 4, 8, 16, 32, 64, respectively. The following is the statistics of the running time of each function. All the data presented in the table are the average value of the program running 10 times.

#### 7.2.2. Performance Analysis of CBOL-Ring Mechanism

There are two experiments in this section, the CBOL-ring mechanism is divided into 13 functions in the test, which belong to the borrower, bank and audit nodes.

The first experiment mainly tests the running time of the functions at each stage. At the registration stage, the setup function mainly spends time on generating the bank’s public and private keys, the borrower’s ID, and the borrower’s maximum loan limit, its running time is irrelevant to the length of the binary string. Suppose a bank has to generate 30 pairs of public and private keys, the running time is 0.0117 s. Therefore, this experiment focuses on testing the relationship between the running time of the remaining 12 functions and the change of the length of the binary string.

Figure 5 shows the running time of the borrower, bank, and audit node respectively, where Figure 5a shows the running time of each function on the borrower side, Figure 5b shows the running time of each function on the bank side, Figure 5c shows the running time of each function on the audit node. It can be found from Table 2 and Figure 5 that when the length of the binary string changes, except that the running time of the function that generates range proofs (GenRproof) and the functions that verify range proofs (VerRproof, VerRp) is changed, the running time of other functions changes little. If we ignore the impact on the system operation, it can be considered that the change in the length of the binary string does not affect the running time of those functions. The reason is that other functions do not need to convert an amount or random number into a binary string when generating the proof data, their running times are not affected by the binary string length. At the same time, it indicates that the running time of the functions to generate the range proof and verifies the range proof change greatly; they are basically linear co-related with the length of the binary string. It is because when generating the range proof, the amount needs to be converted into a binary string, then the signature is generated by using the corresponding random number as the private key and the commitment generated by each bit of the binary string as the public key. The number of the signature is related to the length of the binary string. When the length of the binary string is l, the number of signatures generated is also l, and the number to call the signature function is l. Therefore, the running time of the function that generates the range proof is proportional to the length of the binary string. Similarly, the running time of the function that verifies range proof is also proportional to the length of the binary string.

The second experiment tests the size of the communication volume at each stage. The size of the audit node’s communication volume refers to the number of bytes related to the algorithm in the transaction, and the size of the communication volume of the borrower and the bank are both the size of the data related to the CBOL-ring mechanism. The communication volume between the bank and the borrower in the registration stage is independent of the length of the binary string, it is most affected by the number of banks in the bank union. Assuming that the number of banks in the bank union is 10, the communication volume is 606 bytes. Therefore, this experiment mainly tests the communication volume of the initialization stage, loan stage and repayment stage.

From Table 3, it can be found that the communication volume of the participants in the initialization phase remains basically unchanged. Ignoring the impact of system operation, it can be considered that the communication volume at this stage is not affected by the change of the binary string. This is because in the initialization stage, the borrower does not need to generate the range proof, and only needs to generate the commitment of the maximum loan amount and the knowledge proof of the random number commitment and the point H. These data do not need to be generated by bit, that is, there is no need to convert the amount or the random number into a binary string, so the proof size has nothing to do with the binary length. In addition, according to Table 3, it can be found that in the loan and repayment stage, the communication volume is mainly constituted by the range proof, and its size is mainly affected by the size of the range proof. This is because the number of the range proof is equal to the length l of the binary string, so the size of the range proof is linearly related to the length of the binary string, the communication volume is also basically linearly related to the length of the binary string. In addition, according to Table 3, it can be found that in the loan and repayment stage, the communication volume is mainly composed of the range proof, and its size is mainly affected by the size of the range proof. This is because the number of the range proof is equal to the length l of the binary string, so the size of the range proof is linearly related to the length of the binary string, the size of the communication volume is also linearly related to the length of the binary string.

#### 7.2.3. Comparison between CBOL-Ring Mechanism and CBOL-Bullet

Through the analysis in Section 7.2.2, we can know that the communication volume of each participant is mainly affected by the size of the distance proof, but in the registration phase and initialization phase, the size of the communication volume between each participant is independent of binary string length. At the same time, the range proof size of the bank and audit node is consistent with the borrower. Therefore, in the first experiment, we compares the size of the range proof generated by the borrower in two mechanisms. The range proof includes the range proof of the loan amount and the range proof of the new remaining loanable amount.

It can be found from the data recorded in Table 4 and the change of range proof size shown in Figure 6, the size of the range proof generated by the CBOL-bullet is significantly smaller than the size of the range proof generated by the CBOL-ring mechanism. And as the length of the binary string continues to increase, the gap between the two is getting larger and larger. In CBOL-ring mechanism, the number of elements contained in a single range proof is fixed, but the total number of range proofs is the same as the length of binary string, so the size of range proof is linear with the length of the binary string, that is, when the length of the binary string increases exponentially, the size of the range proof increases exponentially, but in the CBOL-bullet, the number of elements contained in the range proof has a logarithmic relationship with the length of the binary string, so the size of range proof increases logarithmically.

In the second experiment, we compared the time difference of generating range proof and verifying range proof between the CBOL-ring mechanism and the CBOL-bullet. The range proof includes the range proof of the loan amount and the range proof of the new remaining loanable amount.

Through the analysis of Table 5 and Figure 7, it can be found that the time for the borrower to generate the range proof in the CBOL-bullet is significantly higher than that in the CBOL-ring mechanism. This is because, in the CBOL-ring mechanism, the time to generate a range proof for each bit of the binary string is a 6-point multiplication operationsand 2-exponent arithmetic operation, that is, a total range proof requires 6l point multiplication operations and 2l exponent arithmetic operations, but in CBOL-bullet, the time to generate the range proof is 15l + 4logl + 1 point multiplication operation and 12l + logl + 5 exponent arithmetic operations, which is much longer than the time to generate range proof in CBOL-ring mechanism. So the CBOL-bullet uses time in exchange for the optimization of communication volume. The CBOL-bullet is suitable for scenarios with high communication volume requirements and low runing time requirements.

Table 6 and Figure 8 show the time difference of verifying the range proof between the CBOL-ring mechanism and the CBOL-bullet, where Figure 8a is the time difference for the bank to verify range proof and Figure 8b is the time difference for the audit node to verify range proof. It can be found that the running time of the CBOL-ring mechanism and the CBOL-bullet both showed an upward trend, and the running time of the CBOL-bullet is slightly higher than that of the CBOL-ring mechanism, this is because, in the CBOL-ring mechanism, the time to verify the range proof is 5l point multiplications and l exponential arithmetic operations. And in CBOL-bullet, the time to verify the range proof is 5l + 2logl + 6 point multiple operation and 3l + logl + 3 exponential operation, which is slightly longer than the time to generate range proof in CBOL-ring mechanism, but this is within the acceptable range.

## 8. Conclusions

This paper mainly studies the privacy protection of bank data in the process of sharing. In order to achieve the public verifiability of bank privacy data, this paper proposes the CBOL-ring mechanism using commitment, ring signature, encryption, digital signature, and range proof. This mechanism generates the relevant proofs of the privacy data that realize the verifiability of the data without revealing privacy. At the same time, this paper proposes the CBOL-bullet by using the Bulletproofs algorithm, which can effectively reduce the size of the range proof, reduce the communication volume between the participants, and solve the problem of the linear increase in the communication volume in the CBOL-ring mechanism, enabling the CBOL-ring mechanism to be used in scenarios with long binary string lengths. Finally, we test the performance of the two schemes through simulation experiments and compare the results.

## Figures and Tables

**Figure 1 entropy-22-00619-f001:**
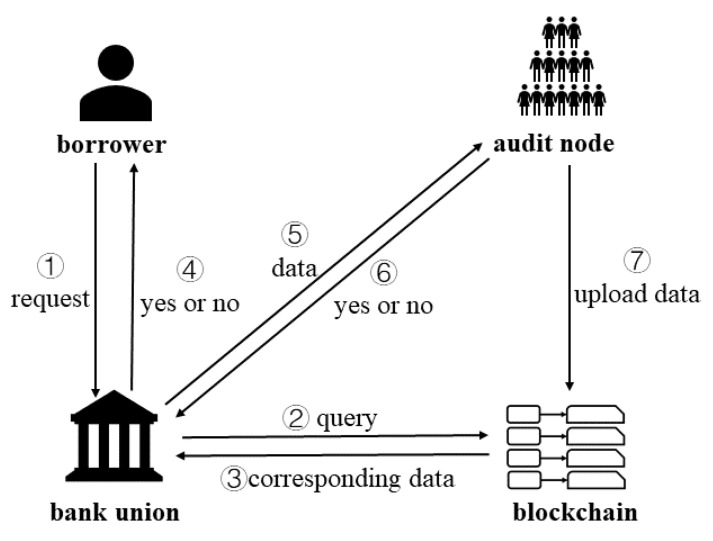
The Framework of the Mechanism Proposed.

**Figure 2 entropy-22-00619-f002:**
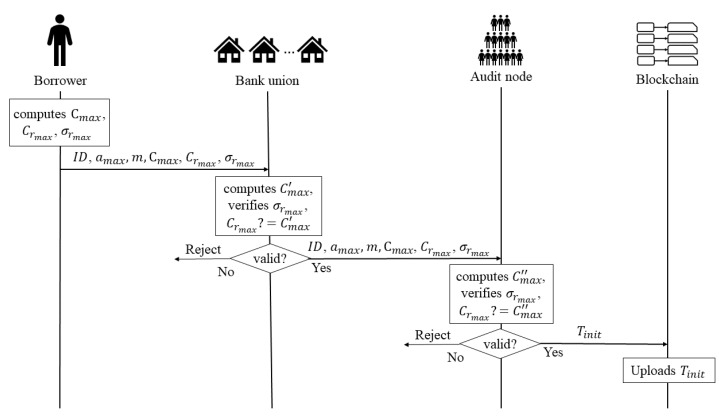
The Process of the Initialization Stage.

**Figure 3 entropy-22-00619-f003:**
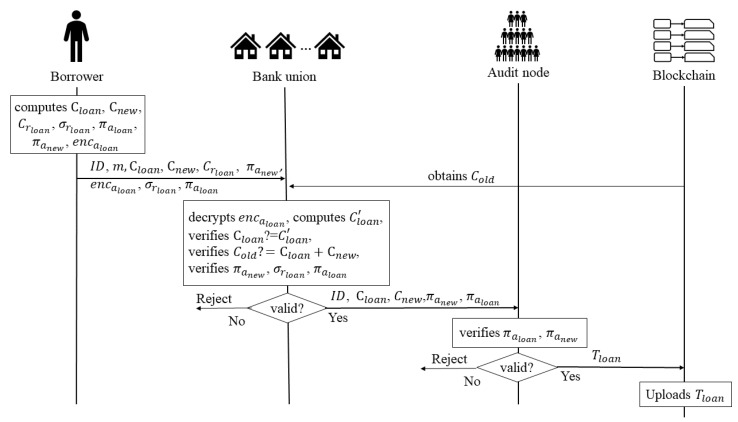
The Process of the Loan Stage.

**Figure 4 entropy-22-00619-f004:**
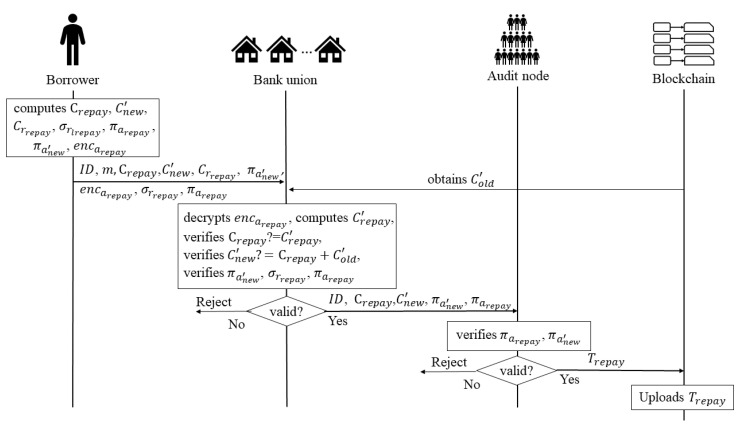
The Process of the Repayment Stage.

**Figure 5 entropy-22-00619-f005:**
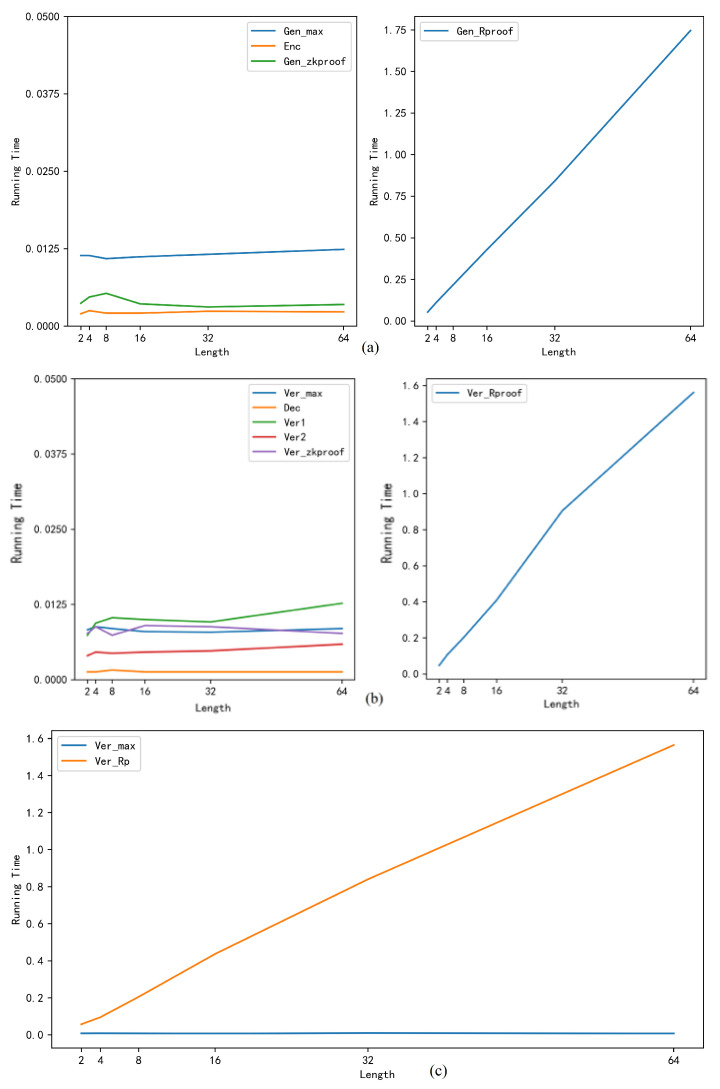
The Running Time of Each Function (s).

**Figure 6 entropy-22-00619-f006:**
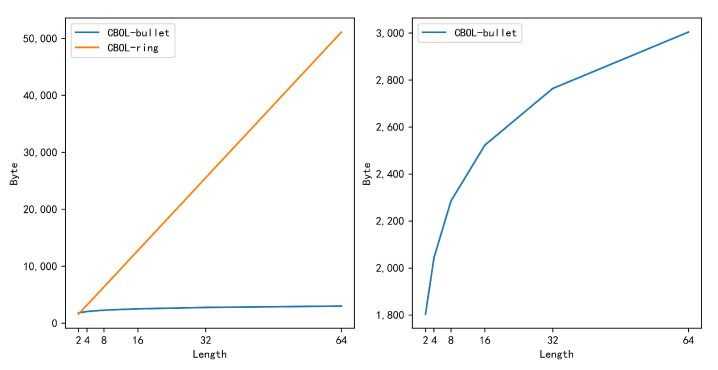
Comparison Range Proof Size of Two Mechanisms (byte).

**Figure 7 entropy-22-00619-f007:**
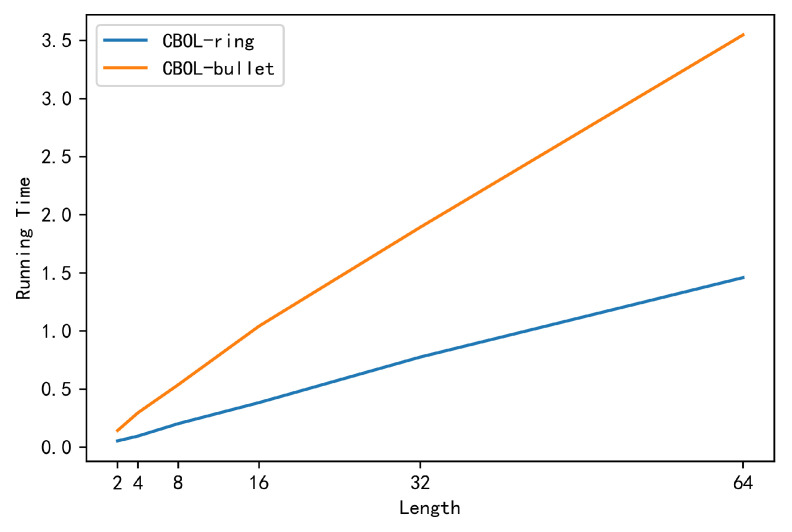
Time for the Borrower to Generate a Rang Proof (s).

**Figure 8 entropy-22-00619-f008:**
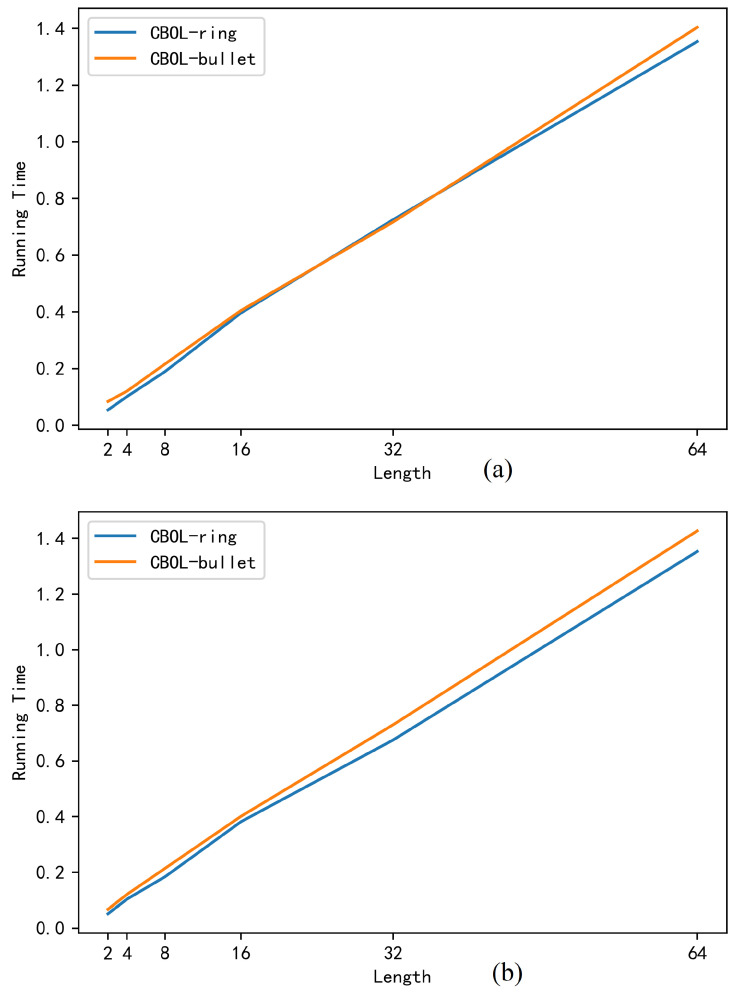
Time to Verify a Rang Proof (s).

**Table 1 entropy-22-00619-t001:** Range Proof Size.

	G	$Z_{p}$
Size	l	5l

**Table 2 entropy-22-00619-t002:** The Running Time of Each Function.

Role	Stage	Running Time (s)					
		2	4	8	16	32	64
Borrower	Genmax	0.0114	0.0114	0.0109	0.0112	0.0116	0.0124
	Enc	0.0020	0.0025	0.0021	0.0021	0.0024	0.0023
	GenRproof	0.0541	0.1113	0.2177	0.4300	0.8422	1.7457
	Genzkproof	0.0037	0.0047	0.0053	0.0036	0.0031	0.0035
Bank	Vermax	0.0083	0.0088	0.0085	0.0080	0.0079	0.0085
	Dec	0.0013	0.0013	0.0016	0.0013	0.0013	0.0013
	Ver1	0.0074	0.0094	0.0103	0.0100	0.0096	0.0127
	Ver2	0.0040	0.0046	0.0044	0.0046	0.0048	0.0059
	VerRproof	0.0476	0.1074	0.2033	0.4097	0.9053	1.5618
	Verzkproof	0.0077	0.0088	0.0074	0.0090	0.0088	0.0077
Audit nodes	VerM	0.0089	0.0092	0.0088	0.0078	0.0101	0.0081
	VerRp	0.0576	0.0958	0.2056	0.4371	0.8402	1.5646

**Table 3 entropy-22-00619-t003:** The Communication Size of Each Stage.

Stage	Role	Communication Size (Byte)						
			**2**	**4**	**8**	**16**	**32**	**64**
initialization	Borrower	range proof	0	0	0	0	0	0
		total communication	393	393	394	395	394	393
	Bank	range proof	0	0	0	0	0	0
		total communication	388	388	389	390	389	388
	Audit node	range proof	0	0	0	0	0	0
		total communication	388	388	389	390	389	388
Loan	Borrower	range proof	1596	3187	6368	12,756	25,547	51,128
		total communication	2121	3712	6892	13,281	26,072	51,653
	Bank	range proof	1596	3187	6368	12,756	25,547	51,128
		total communication	1597	3188	6869	12,757	25,548	51,129
	Audit node	range proof	1596	3187	6368	12,756	25,547	51,128
		total communication	1597	3188	6869	12,757	25,548	51,129
Repayment	Borrower	range proof	1602	3192	6375	12,763	25,549	51,129
		total communication	2127	3717	6899	13,287	26,074	51,654
	Bank	range proof	1602	3192	6375	12,763	25,549	51,129
		total communication	1603	3193	6376	12,764	25,550	51,130
	Audit node	range proof	1602	3192	6375	12,763	25,549	51,129
		total communication	1603	3193	6376	12,764	25,550	51,130

**Table 4 entropy-22-00619-t004:** Comparison Range Proof Size of Two Mechanisms.

Mechanism	Range Proof Size (Byte)					
	2	4	8	16	32	64
CBOL-ring	1597	3188	6370	12,757	25,544	51,126
CBOLP-bullet	1804	2044	2285	2523	2764	3004

**Table 5 entropy-22-00619-t005:** Time for the Borrower to Generate a Proof of Scope.

Mechanism	Running Time (s)					
	2	4	8	16	32	64
CBOL-ring	0.0539	0.0951	0.2024	0.3830	0.7755	1.4591
CBOL-bullet	0.1437	0.2953	0.5371	1.0406	1.8926	3.5480

**Table 6 entropy-22-00619-t006:** Time to Verify a Proof of Scope.

Role	Mechanism	Running Time (s)					
		2	4	8	16	32	64
Bank	CBOL-ring	0.0544	0.1017	0.1897	0.3967	0.7254	1.3535
	CBOL-bullet	0.0845	0.1209	0.2165	0.4053	0.7171	1.4038
Audit node	CBOL-ring	0.0519	0.1038	0.1841	0.3819	0.6755	1.3530
	CBOL-bullet	0.0678	0.1207	0.2141	0.4014	0.7302	1.4270

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
