# Peer review of "CBOL: Cross-Bank Over-Loan Prevention, Revisited"

_entropy, 2020, doi:10.3390/e22060619_

Round 1

Reviewer 1 Report

The paper introduces blockchain-based across-bank over-loan prevention mechanism - a privacy mechanism in banking.

The paper requires extensive organizational, writing, grammatical corrections (for instance, too many long sentences with grammatical mistake and difficult to read, convert them to short and readable sentence)

Demonstrate the techniques using flowchart/diagrams (difficult to follow with so many equations, present them first in simpler form and then mathematical form)

Diagrams should use the method name instead of mechanism 1 and mechanism 2 with more explanation of results.

Try to use more recent references.

Author Response

Dear Reviewer,

Thank you very much for your efforts in helping us to improve the quality of this paper. We would like to reply your valuable comments point-by-point.

Best regards.

Xiaoya Hu.

Reviewer 2 Report

This paper is interesting and has provided sound solutions for existing problems. But there are also several common approaches. So the challenge is to convince others why authors' approach maybe the most relevant, and not other types. Comparisons with other approaches or similar approaches should be included to some extent. Those comparisons will need references.

Apart from the fact blockchain papers are more likely to be considered, the novelty should be explained well too. the issues caused by the financial crisis and also the subsequent followed-up can be further expanded. 

1) To guide authors, some references are as follows.

a) Financial crisis means risk management, analysis and monitoring are important. New ways to achieve that, together with a framework that can work with financial services, scientists and the public, there are some examples, including 1 with full validation and details.

Toward Business Integrity Modeling and Analysis Framework for Risk Measurement and Analysis. Applied Sciences. 2020, 10(9), 3145.

b) Identify management is another alternative approach that can divide different users' roles, permission and privilege similar to the authors' objectives. A comparison of functionality and other aspects can be described.

On orchestrating service function chains in 5G mobile network. IEEE Access. 2019, 7, 39402-39416.

c) Security solutions should work in different domains. Robust solutions matter. In this case, it can be generic to financial services with strong algorithms and advanced encryption dealing with large data and number of users. Compare to such approaches, what are things new in the authors' approach and also what lessons can be learned for cross-disciplined usage scenarios?

Privacy-preserving smart IoT-based healthcare big data storage and self-adaptive access control system. Information Sciences, 2019, 479, 567-592.

2) Please justify research contributions. Please also get proper proofreading for this paper.

Author Response

Dear Reviewer,

Thank you very much for your efforts in helping us to improve the quality of this paper. We would like to reply your valuable comments point-by-point.

Best regards,

Xiaoya Hu.

Round 2

Reviewer 1 Report

Properly addressed the comments.

Reviewer 2 Report

Accept.

This manuscript is a resubmission of an earlier submission. The following is a list of the peer review reports and author responses from that submission.